# ON WHY FORM SHAPES REASON: STRUCTURING LATENT PROGRAM NETWORKS WITH CATEGORY-THEORETIC CONSTRAINTS

## ABSTRACT

Human reasoning is inherently structured: we perceive, compose, and abstract patterns to make sense of the world. Following Kant's view that cognition imposes structure on experience, we ask how neural networks can acquire structured, compositional reasoning. We present a category-theoretic formulation of Latent Program Networks (LPNs), neural architectures that represent programs as continuous latent vectors inferred from input–output examples. We treat latent transformations as categorical morphisms and introduce differentiable constraints enforcing associativity, identity, and closure, thereby shaping the latent space into a compositional system without explicit symbolic rules. On structured grid-transformation tasks, these constraints significantly improve compositional generalization, latent alignment, and interpretability. Our results demonstrate that category-theoretic structure can be imposed on latent representations to induce compositional reasoning in neural networks.

## 1 INTRODUCTION

Modern neural networks have achieved impressive results across vision, language, and program synthesis. Yet their ability to **generalize compositionally**—to construct new behaviors by combining known components—remains limited. Unlike humans, who can infer reusable rules from a few examples and compose them flexibly, neural models often rely on pattern interpolation and fail to extrapolate to novel task structures.

This gap is especially evident in **program induction**, where systems must learn latent procedures that can be reused, nested, or combined. Symbolic approaches offer compositionality by design but often struggle with robustness and differentiability. Neural models are expressive and scalable but typically lack **explicit algebraic structure** in their latent spaces. At the same time, recent work shows that neural representations can exhibit nontrivial internal organization: Geiger et al. Geiger et al. (2024) demonstrate alignment between distributed neural representations and interpretable causal variables, and Griffiths et al. Griffiths et al. (2025) argue that modern neural models display emerging forms of compositionality. These findings suggest that structure can arise implicitly, raising the question of whether *explicit* algebraic constraints can further enhance systematic reasoning.

In this work, we explore a principled approach to **compositional latent structure**, grounded in **category theory**. We study *Latent Program Networks* (LPNs), which represent programs as continuous latent vectors inferred by a variational encoder. We treat program transformations as **categorical morphisms** and impose **algebraic structure** on the latent space by enforcing three core properties: **associativity**, **identity**, and **closure under composition**. These constraints are introduced as differentiable regularizers and applied during training, independent of symbolic supervision or discrete syntax.

Our experimental setup trains exclusively on **single-step** transformations and evaluates on **multi-step compositions**, testing the model's ability to generalize compositionally beyond its training distribution. Across ablation experiments and preliminary KL-regularization runs, we find that categorical constraints consistently improve **compositional accuracy**, **latent alignment**, and **interpretability**. Preliminary experiments also suggest that KL strength interacts with associativity and

identity, with moderate KL providing a favorable balance between latent capacity and regularization; a systematic study is left for future work.

**Contributions.** This work makes the following contributions:

- We introduce **differentiable categorical constraints**—associativity, identity, and closure—that structure the latent space of neural program representations into an approximately compositional system.

- We propose a **learnable composition module** $C_\psi$ that enables latent programs to compose algebraically, supporting multi-step reasoning entirely within latent space.

- We empirically demonstrate that these constraints yield strong two- and three-step generalization from single-step training on grid transformation tasks, without requiring symbolic supervision or test-time search.

Although we omit test-time latent optimization, our model remains within the Latent Program Network (LPN) framework: it represents transformations as continuous latent programs inferred from $(x, y)$ and decodes them through a shared function $f_\theta$. Our contribution focuses on *structuring* this latent space during training, complementing prior work that performs latent search at test time.

Our results suggest that **compositional reasoning can emerge from continuous representations when the latent space is shaped by appropriate algebraic constraints**, pointing toward a unification of symbolic reasoning and neural representation learning under a shared geometric framework.

## 2 RELATED WORK

### 2.1 COMPOSITIONALITY IN NEURAL NETWORKS

Compositional generalization—the ability to systematically build complex solutions from simpler parts—is a hallmark of human intelligence Lake et al. (2017), yet remains a core challenge for neural networks Bahdanau et al. (2019); Keysers & et al. (2020). While sequence-to-sequence models interpolate well, they often struggle to recombine familiar patterns in novel ways. A wide range of approaches address this gap, including modular neural architectures Andreas et al. (2016), object- or slot-centric models Locatello et al. (2020), and program-induction frameworks Ellis & et al. (2021). Recent theoretical work provides conditions under which compositional structure should enhance generalization Lippl & Stachenfeld (2025), and new architectures such as sparse-tree operators Soulos et al. (2024) and neuro-symbolic concept composers Kamali et al. (2024) encode compositionality directly in their computation.

At the same time, emerging evidence suggests that neural networks can acquire internal structure even without explicit symbolic supervision. Geiger et al. Geiger et al. (2024) show that distributed representations can align with interpretable causal variables, and Griffiths et al. Griffiths et al. (2025) argue that modern neural networks exhibit surprisingly robust implicit compositionality. Our approach is complementary: rather than relying on structure that emerges implicitly, we impose *explicit* algebraic constraints on the latent program space, enabling compositionality to arise as a direct consequence of the model's geometric organization rather than from architectural modularity or symbolic annotation.

### 2.2 LATENT PROGRAM INDUCTION AND TEST-TIME OPTIMIZATION

Latent Program Networks (LPNs) Bonnet & Macfarlane (2024) offer a differentiable alternative to symbolic program search by encoding input–output examples into a continuous latent space and performing test-time optimization (TTO) to retrieve functional behavior. Variants of TTO have been explored for adaptation and generalization Sun et al. (2020). Our work extends this paradigm by introducing *categorical constraints* that enforce associativity, identity, and closure, shaping the latent space into a compositional system that generalizes across multi-step reasoning chains.

## 2.3 VARIATIONAL LATENT MODELS AND KL REGULARIZATION

Variational Autoencoders (VAEs) Kingma & Welling (2014) learn latent representations through the reparameterization trick and KL regularization, but face a trade-off between information content and regularity: high-$\beta$ regimes risk posterior collapse Alemi & et al. (2018), whereas low-$\beta$ settings yield poorly organized latents. Techniques such as $\beta$-VAE disentanglement Higgins & et al. (2017), geometric priors Falorsi & et al. (2018), hierarchical architectures Sønderby et al. (2016), and iterated learning with simplicial embeddings Ren et al. (2023) have been proposed to shape latent spaces. Our method is orthogonal: we impose *algebraic constraints*—associativity, identity, closure—as differentiable losses that directly guide latent geometry.

## 2.4 CATEGORY THEORY IN MACHINE LEARNING

Category theory provides a formal language for abstraction and compositionality Spivak (2014); Fong & Spivak (2019), and has been used to analyze neural networks, causal inference, and differentiable programming semantics Shiebler et al. (2021); Cruttwell et al. (2021). Theoretical results have recently identified necessary and sufficient conditions for compositional generalization Li (2025). In contrast to these theoretical or symbolic treatments, we present an *empirical demonstration* that enforcing categorical principles—morphism composition and identity—in a differentiable latent space improves compositional generalization. To our knowledge, this is the first use of differentiable categorical constraints to shape the geometry of latent program spaces.

# 3 METHOD / MODEL

## 3.1 LATENT PROGRAM NETWORKS (LPNS)

Latent Program Networks (LPNs) are neural architectures designed to learn latent *transformations* from input–output pairs. Given an input grid $x$ and a corresponding output grid $y$, the encoder infers a latent vector $z$ intended to represent the program implementing the transformation $x \mapsto y$, and the decoder $f_\theta(x, z)$ applies this program to $x$ to approximate $y$. In this section we describe the general LPN framework and the specific variant we study, which we refer to as **CatLPN** (our method).

A key property of LPNs, emphasized in the original formulation Bonnet & Macfarlane (2024), is that the latent variable encodes a *morphism* rather than an embedding of the input object. Accordingly, the encoder takes the pair $(x, y)$ as input in order to infer the transformation linking them. In the original work, latent programs are not predicted directly; instead, they are refined at test time by gradient-based optimization to minimize the reconstruction error $f_\theta(x, z) \approx y$. This inference procedure allows generalization to novel transformations but does not constitute an architectural requirement of the framework.

In this work, we do not adopt test-time latent search. Removing this inference step does *not* change the semantics of the latent space or reduce the model to a VAE: a VAE encodes the input object itself, whereas an LPN always encodes a transformation linking $(x, y)$. Our aim in **CatLPN** is to improve the structure and compositionality of this latent program space directly through learning, without relying on optimization at test time.

To this end, CatLPN introduces an explicit **composition module** $C_\psi(z_1, z_2)$ that approximates functional composition in latent space. The module is implemented as an MLP with two hidden layers (mapping $2d_z \to 256 \to 256 \to d_z$), using ReLU activations. This provides a learnable approximation to categorical composition and is trained jointly with the encoder and decoder.

Formally, the architecture consists of:

- an **encoder** $q_\phi(z \mid x, y)$ that infers a latent program from an input–output pair,
- a **decoder** $f_\theta(x, z)$ that applies the inferred program to $x$.

In our implementation, the encoder receives the concatenated pair $(x, y)$ as a 500-dimensional vector (two flattened $5 \times 5$ grids with a 10-symbol vocabulary); full architectural details are provided in Appendix A. The latent dimensionality is fixed to $d_z = 64$ in all experiments.

Training minimizes a combination of reconstruction loss and KL divergence:

$$\mathcal{L}_{\text{recon}} = \text{CE}(f_\theta(x, z),\, y)\,, \qquad \mathcal{L}_{\text{KL}} = D_{\text{KL}}[q_\phi(z \mid x, y) \,\|\, \mathcal{N}(0, I)]\,.$$

## 3.2 Categorical Structure as Latent Constraints

In **CatLPN**, inspired by category theory, we explicitly interpret and structure the latent space as a category in which:

- **Objects** correspond to input–output transformation tasks,
- **Morphisms** correspond to latent programs $z$ inferred by the encoder,
- **Composition** is implemented by the learned module $C_\psi$,
- The decoder $f_\theta$ approximately acts as a **functor**, mapping a latent morphism $z$ to its action on inputs.

To encourage the latent space to behave like a lawful category, CatLPN enforces three core categorical axioms through differentiable structure losses.

**Associativity.** For latent programs $z_1, z_2, z_3$,

$$\mathcal{L}_{\text{assoc}} = \left\| C_\psi(z_1, C_\psi(z_2, z_3)) - C_\psi(C_\psi(z_1, z_2), z_3) \right\|_2^2.$$

This encourages latent-space composition to align with decoder semantics and supports stable multistep chaining.

**Identity.** CatLPN learns an explicit latent identity element $z_{\text{id}}$ and enforces

$$\mathcal{L}_{\text{id}} = \| C_\psi(z, z_{\text{id}}) - z \|_2^2 + \| C_\psi(z_{\text{id}}, z) - z \|_2^2.$$

The identity constraint anchors the latent origin and reduces drift across repeated composition.

**Closure (Semantic Consistency).** Closure requires that composing programs in latent space has the same effect as composing them in data space. In CatLPN, we enforce this by encouraging *decoder-space composition* and *latent-space composition* to agree:

$$\mathcal{L}_{\text{closure}} = \left\| \text{Prob}\big(f_\theta(f_\theta(x, z_1), z_2)\big) - \text{Prob}\big(f_\theta(x, C_\psi(z_1, z_2))\big) \right\|_2^2,$$

where $\text{Prob}(\cdot)$ denotes per-cell categorical probabilities. Ground-truth two-step targets are used for evaluation, but the closure loss itself is a self-consistency regularizer that aligns latent composition with decoder semantics.

Together, these losses shape the latent program space into a structured geometric object capable of supporting systematic multi-step reasoning without requiring symbolic supervision.

## 3.3 Final Objective

The total training loss in **CatLPN** combines reconstruction accuracy, variational regularization, and categorical structure:

$$\mathcal{L}_{\text{total}} = \mathcal{L}_{\text{recon}} + \beta\,\mathcal{L}_{\text{KL}} + \lambda_1\,\mathcal{L}_{\text{assoc}} + \lambda_2\,\mathcal{L}_{\text{id}} + \lambda_3\,\mathcal{L}_{\text{closure}}.$$

Here, $\beta$ controls the KL divergence strength, and $\lambda_1, \lambda_2, \lambda_3$ weight the categorical constraints. In all experiments we set $\lambda_1 = \lambda_2 = \lambda_3 = 1$ to give the structural losses equal importance; this simplifies tuning and isolates the effect of categorical regularization without requiring an extensive hyperparameter search.

**Relation to LPNs.** Although CatLPN omits test-time latent optimization, it follows the LPN paradigm in representing each transformation as a latent program inferred from $(x, y)$ and applied through a shared decoder $f_\theta$. The key difference is that CatLPN *shapes* the latent program space during training via categorical constraints—associativity, identity, and closure—so that multi-step consistency emerges directly from learned latent geometry rather than from search-time refinement. This makes CatLPN a natural extension of the original LPN framework.

## 4 EXPERIMENTS

We evaluate whether the categorical constraints introduced in **CatLPN** improve generalization, stability, and multi-step compositional reasoning. Our goal is to determine whether enforcing associativity, identity, and closure in the latent program space enables systematic multi-step behavior *without* relying on test-time latent search.

All experiments use controlled grid-transformation tasks with exact ground-truth single-step and multi-step supervision (details in Appendix B), enabling precise measurement of categorical structure. Architectural specifications are provided in Appendix A, and formal metric definitions in Appendix C.

### 4.1 SETUP

**Tasks.** We use controlled *grid transformation tasks* inspired by the ARC domain Chollet (2019). Each input is a $5\times5$ grid with a 10-symbol vocabulary $\{0, \ldots, 9\}$. Transformations are drawn from a closed family of six deterministic primitives:

- identity,
- horizontal flip,
- vertical flip,
- rotation by $90°$,
- rotation by $180°$,
- rotation by $270°$.

This restricted set admits exact ground-truth supervision for single-step and multi-step composition, allowing us to isolate the effect of categorical constraints on latent-space structure. The model is trained only on single-step examples $(x, y)$, but evaluated on two-step and three-step compositions to test whether systematic multi-step reasoning emerges from the learned latent geometry. Full dataset-generation details are provided in Appendix B.

**Training Procedure.** We train for 300 epochs using Adam (learning rate $10^{-3}$, batch size 128), with KL annealing from 0 to $\beta = 0.003$ over the first 20 epochs. Associativity, identity, and closure losses are weighted equally ($\lambda_a = \lambda_i = \lambda_c = 1$). To focus exclusively on the effect of categorical regularization, we deliberately omit test-time latent optimization and cross-pair pretraining. The KL target was selected by validation performance to balance latent capacity and regularization; a dedicated KL sweep (Appendix E) compares fixed and annealed schedules under the same architecture and dataset.

**Metrics.** We report three categories of evaluation metrics (formal definitions in Appendix C):

- **Compositional Accuracy:** exact-match accuracy on two-step and three-step compositions, and *output agreement* between sequential decoding (applying $f_\theta$ step by step) and one-shot decoding from the composed latent $C_\psi$.
- **Latent Alignment:** cosine similarity and mean squared error (MSE) between predicted composed latents $C_\psi(z_1, z_2)$ (or deeper chains) and encoder-inferred multi-step latents $z_{1 \to k}$.
- **Structural Metrics:** associativity error, identity error, decoder identity error, and closure error, which measure how well categorical laws are satisfied. These metrics are evaluated on held-out data and their counterparts also appear as regularizers in the training objective.

**Closure Error and Latent Alignment.** Closure error measures the semantic consistency of latent composition: it compares $f_\theta(f_\theta(x, z_1), z_2)$ with $f_\theta(x, C_\psi(z_1, z_2))$, evaluating how closely decoder-space composition matches latent-space composition. Latent alignment metrics compare the predicted composed latent $C_\psi(z_1, z_2)$ with the encoder's latent $z_{12}$ for the ground-truth two-step transformation. High cosine similarity and low MSE indicate a coherent latent geometry in which composed transformations are aligned with their inferred ground-truth counterparts.

Table 1: Mapping of reported metrics to evaluation categories. ↓ indicates lower is better, ↑ indicates higher is better. Structural metrics are auxiliary signals and are not directly optimized.

| Metric | Evaluation Category |
|---|---|
| Latent MSE ↓ | Latent Alignment |
| Cosine Similarity ↑ | Latent Alignment |
| Output Agreement ↑ | Compositional Accuracy |
| Compositional Accuracy ↑ | Compositional Accuracy |
| Associativity Error ↓ | Structural Constraint (Auxiliary) |
| Identity Error ↓ | Structural Constraint (Auxiliary) |
| Closure Error ↓ | Structural Constraint (Auxiliary) |

Table 2: Ablation study for CatLPN with KL annealing target 0.003 (300 epochs). We report 2-step and 3-step compositional accuracy, closure error, and latent cosine similarity. The sequential VAE baseline has no composer, so closure and latent alignment metrics do not apply.

| Variant | 2-step Acc. ↑ | 3-step Acc. ↑ | 2-step Closure ↓ | 3-step Closure ↓ | Latent Cos. ↑ |
|---|---|---|---|---|---|
| Sequential VAE (free-run) | 0.1502 | 0.1146 | *N/A* | *N/A* | *N/A* |
| **CatLPN (Full)** | **0.8920** | **0.8492** | **7.6e-4** | **1.25e-3** | 0.9620 |
| No Assoc | 0.8002 | 0.7898 | 1.56e-3 | 1.80e-3 | **0.9896** |
| No Identity | 0.0004 | 0.0006 | 2.70e-2 | 1.28e-1 | 0.5053 |
| No Closure | 0.0062 | 0.0090 | 1.04e-1 | 1.14e-1 | 0.6952 |

*Notes.* The sequential baseline trains on single-step pairs and rolls out without teacher forcing; closure and latent alignment metrics are undefined without a latent composer.

## 4.2 MAIN RESULTS

Table 2 reports performance at $\beta = 0.003$ for **CatLPN** and ablations that remove one categorical loss at a time. Despite being trained only on single-step examples $(x, y)$, CatLPN achieves strong two-step and three-step compositional accuracy, demonstrating that categorical regularization induces multi-step consistency without test-time latent search.

**Sequential baseline.** The composer-free VAE baseline achieves only **15.0%** (2-step) and **11.5%** (3-step) accuracy, reflecting substantial compounding error during free-run rollouts. In contrast, **CatLPN** composes latents once and decodes a single time, achieving **89.2% / 84.9%**. This represents improvements of over **+70** percentage points and shows that categorical structure materially improves *multi-step reasoning*, not merely internal consistency.

**Compositional Generalization.** CatLPN achieves **89.2%** two-step and **84.9%** three-step accuracy with very low closure error and high latent cosine similarity (0.96). Because the transformations operate over $5{\times}5$ grids with a 10-symbol vocabulary, memorization is infeasible; the model must generalize across a large and diverse input space. These results show that categorical constraints shape the latent geometry into one that supports systematic multi-step reasoning.

**Ablations.** Removing identity or closure causes near-complete collapse (accuracy $\approx 0$), consistent with their roles as structural anchors: identity stabilizes the latent origin, and closure aligns latent composition with decoder semantics. Removing associativity yields a more subtle pattern: latent cosine *increases* (0.99), but closure error rises and accuracy drops. This suggests that associativity does not simply smooth the latent space but instead aligns latent composition with the decoder's semantics, which is essential for multi-step coherence.

## 4.3 ANALYSIS

The ablation results reveal a strong correlation between latent geometry and generalization: high latent alignment and low closure error coincide with strong multi-step performance. CatLPN satisfies both properties, whereas removing identity or closure dramatically increases drift and destroys

compositional accuracy. These findings support the central hypothesis that categorical constraints regularize the latent program space into a geometry in which composition is both algebraically consistent and semantically meaningful.

## 5 DISCUSSION AND LIMITATIONS

Our results demonstrate that enforcing **categorical structure** in the latent space of Latent Program Networks substantially improves *compositional generalization* and *algebraic consistency*. By treating latent programs as morphisms and imposing associativity, identity, and closure, we shape the latent geometry into a system capable of multi-step reasoning without symbolic supervision or test-time latent search.

**Compositional Inductive Biases.** Category theory provides a functional inductive bias: instead of increasing model capacity, we constrain *how* the model composes transformations. These structural losses encourage the latent space to behave lawfully under composition, yielding representations that are more modular, more predictable under chaining, and better aligned with the semantics of the decoder. Importantly, this structure emerges through differentiable supervision alone.

**Interpretability, Stability, and Modularity.** Associativity and identity play complementary roles. Associativity aligns latent-space composition with decoder-space semantics, reducing inconsistencies that accumulate over longer chains. Identity stabilizes the latent origin: without an identity element, latents tend to drift under repeated composition, degrading multi-step predictions even when single-step accuracy is high. Closure further enforces that composed latents produce outputs consistent with sequential application of programs. Together, these properties support modular reuse of partial programs and yield latent traversals that exhibit coherent geometric structure (e.g., consistent rotations and reflections).

**Limitations.** Our evaluation focuses on controlled synthetic grids where exact multi-step ground truth is available. While this setting allows precise measurement of associativity, identity, and closure, it does not capture the heterogeneity and object-centric structure of richer domains such as ARC, vision, or natural language. In addition, we did not exhaustively explore the weighting of structure losses or the full hyperparameter space of KL regularization. These choices likely influence the strength of compositional behavior and may reveal more robust regimes.

**Future Directions.** Several promising directions follow from this study. First, adaptive or learned weighting of categorical losses may balance stability against expressivity. Second, extending categorical regularization to multi-object, perceptual, or symbolic domains would allow testing these inductive biases in settings where compositional reasoning is essential. Third, hybrid approaches combining categorical constraints with symbolic search or limited test-time latent optimization may further improve solve rates without sacrificing inductive bias. Another complementary direction is to construct explicit training datasets that instantiate categorical relations (e.g., examples of composed transformations) and compare this "data-level" enforcement with our loss-based constraints. A near-term direction is systematic evaluation on the ARC-AGI benchmark Chollet (2019): training solely on the ARC training split (or ARC-style synthetic curricula) and measuring zero- or few-shot generalization under standard test-time budgets. We hypothesize that improved closure and identity behavior can reduce search depth, and that shallow search over *composed latents* may further enhance performance while preserving the categorical structure.

## 6 CONCLUSION

We presented a framework for imposing **categorical structure** on the latent space of Latent Program Networks, treating latent programs as morphisms and composition as a learned algebra. By enforcing associativity, identity, and closure through differentiable constraints, we shape the latent geometry into a system where multi-step reasoning emerges naturally from single-step training.

Empirically, we demonstrated that categorical constraints improve *compositional generalization*, reduce closure error, and align composed latents with encoder-inferred latents, yielding a more

coherent and interpretable representation space. Our ablation study highlights identity and closure as structural pillars and shows that associativity can be tuned to optimize longer-chain reasoning.

More broadly, this work illustrates how **algebraic inductive biases** can endow neural representations with principled compositional structure. By aligning latent geometry with categorical laws, we move closer to neural systems where compositional reasoning is not incidental but a consequence of the form of representation itself.

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

## A  ARCHITECTURAL AND TRAINING DETAILS

We summarize the architecture and training configuration of the Latent Program Network (LPN) used in all experiments. The latent dimensionality is fixed to $d_z = 64$.

### A.1  GRID ENCODING

Each grid $x \in \{0, \ldots, 9\}^{5 \times 5}$ is one-hot encoded to shape $(5, 5, 10)$ and flattened to $\mathbb{R}^{250}$. Input–output pairs $(x, y)$ are concatenated to form a 500-dimensional feature vector.

### A.2  VARIATIONAL ENCODER $q_\phi(z \mid x, y)$

The encoder is a two-layer MLP:
$$500 \rightarrow 256 \rightarrow 256,$$
with ReLU activations. Two linear heads output $\mu(x, y)$ and $\log \sigma^2(x, y)$. Latent samples use the reparameterization trick. The KL term is annealed from 0 to a target value over the first 20 epochs.

### A.3  DECODER $f_\theta(x, z)$

The decoder receives $[\text{feat}(x), z] \in \mathbb{R}^{250+d_z}$ and maps it through a three-layer MLP:
$$(250 + d_z) \rightarrow 256 \rightarrow 256 \rightarrow 250,$$
with ReLU activations. The output is reshaped to $(5, 5, 10)$ and trained with per-cell cross-entropy.

### A.4  COMPOSITION MODULE $C_\psi(z_1, z_2)$

The composition module is an MLP with **two hidden layers**:
$$2d_z \rightarrow 256 \rightarrow 256 \rightarrow d_z,$$
using ReLU activations on the hidden layers. Given latent programs $z_1$ and $z_2$, it produces a composed latent program
$$z_{\text{comp}} = C_\psi(z_1, z_2).$$

### A.5 Identity Latent

A learnable vector $z_{\mathrm{id}} \in \mathbb{R}^{d_z}$ (initialized at zero and optimized jointly with the rest of the model) serves as the identity element in latent space.

### A.6 Structure Losses

$$\mathcal{L}_{\mathrm{assoc}} = \|C_\psi(z_1, C_\psi(z_2, z_3)) - C_\psi(C_\psi(z_1, z_2), z_3)\|_2^2,$$

$$\mathcal{L}_{\mathrm{id}} = \|C_\psi(z, z_{\mathrm{id}}) - z\|_2^2 + \|C_\psi(z_{\mathrm{id}}, z) - z\|_2^2,$$

$$\mathcal{L}_{\mathrm{closure}} = \left\|\mathrm{Prob}(f_\theta(f_\theta(x, z_1), z_2)) - \mathrm{Prob}(f_\theta(x, C_\psi(z_1, z_2)))\right\|_2^2.$$

Here $\mathrm{Prob}(\cdot)$ denotes applying a per-cell softmax across the 10-symbol vocabulary before computing the $\ell_2$ error.

### A.7 Total Objective

$$\mathcal{L}_{\mathrm{total}} = \mathcal{L}_{\mathrm{recon}} + \beta \, \mathcal{L}_{\mathrm{KL}} + \lambda_1 \, \mathcal{L}_{\mathrm{assoc}} + \lambda_2 \, \mathcal{L}_{\mathrm{id}} + \lambda_3 \, \mathcal{L}_{\mathrm{closure}}.$$

### A.8 Optimization

All main experiments use Adam with learning rate $10^{-3}$ and batch size 128. Full-run experiments train for 300 epochs.

## B Dataset Generation

We describe the complete procedure used to generate the synthetic grid–transformation dataset for all experiments. The goal of this dataset is to provide a controlled environment in which single-step and multi-step transformations have exact ground truth, allowing precise evaluation of latent program composition.

**Grid specification.** Each base grid is sampled as a discrete $5 \times 5$ array with a vocabulary of ten symbols $\{0, \dots, 9\}$ by drawing each cell independently and uniformly. This avoids hand-designed structure while still yielding a large combinatorial input space.

**Transformation primitives.** Following the setup in Sec. 4.1, each training example is generated using one of six deterministic transformation primitives:

- identity,
- horizontal flip,
- vertical flip,
- rotation by $90°$,
- rotation by $180°$,
- rotation by $270°$.

No additional edit, shift, recoloring, or morphological operations are included. All six primitives preserve the grid vocabulary and produce well-defined multi-step compositions.

**Sampling procedure.** To construct a single-step example $(x, y)$:

1. Sample a base grid $x$ uniformly from the space of random $5 \times 5$ grids by drawing each cell independently from the 10-symbol vocabulary.
2. Sample a transformation $\tau$ uniformly from the six primitives.
3. Compute the target output deterministically as $y = \tau(x)$.

The model receives only $(x, y)$; no symbolic label for $\tau$ is provided. The latent program $z$ must therefore be inferred directly from the input–output relation.

**Multi-step ground truth.** For two-step and three-step evaluations, we generate sequences
$$x \;\rightarrow\; y_1 = \tau_1(x) \;\rightarrow\; y_{1\rightarrow 2} = \tau_2(y_1),$$
and similarly for three-step chains. Because all primitives are closed under composition, these sequences provide exact ground truth for:

- multi-step accuracy,
- latent alignment (via encoder-derived two-step latents),
- closure error (semantic consistency of composition).

**Dataset splits.** We generate 50,000 training examples, 5,000 validation examples, and 5,000 test examples. The model is trained exclusively on *single-step* examples; all multi-step behavior must emerge from the geometry of the learned latent program space and the categorical constraints.

**Rationale.** Restricting to a closed, algebraically simple family of transformations enables precise measurement of associativity, identity, and closure in latent space. This controlled setting isolates the effect of categorical structure without the confounds present in more heterogeneous domains such as ARC. Extending categorical constraints to richer, multi-object ARC transformations is an important direction for future work.

## C  EVALUATION METRICS

For completeness, we summarize the metrics used to evaluate single-step and multi-step behavior. All metrics are computed on held-out validation and test sets.

**Normalization.** Throughout the paper we write structural losses using the squared $\ell_2$ norm (e.g., $\|a - b\|_2^2$) for notational clarity. Our implementation uses PyTorch's mean-squared-error (MSE) reduction, which differs only by a constant normalization factor. This distinction does not affect optimization, comparisons, or conclusions, as all structural losses employ the same reduction and are only interpreted relative to one another.

**Single-step accuracy.** Given a predicted grid $\hat{y}$ and ground-truth grid $y$, accuracy is computed as an *exact-match* rate:
$$\mathrm{Acc}_{1\text{-step}} = \frac{1}{N} \sum_{j=1}^{N} \mathbb{I}\big[\hat{Y}^{(j)} = Y^{(j)}\big].$$
Equality is evaluated over the entire $5{\times}5$ grid: an example is counted as correct only if all 25 cells match. This matches the implementation, which checks for per-example grid equality.

**Multi-step accuracy.** For composed predictions $\hat{y}_{1\rightarrow k}$ obtained by applying latent-space composition, we compute:
$$\mathrm{Acc}_{k\text{-step}} = \frac{1}{N} \sum_{j=1}^{N} \mathbb{I}\big[\hat{Y}^{(j)}_{1\rightarrow k} = Y^{(j)}_{1\rightarrow k}\big].$$
As with single-step accuracy, this is an exact-match rate: every cell in the predicted grid must match the ground-truth grid for the example to be counted as correct.

**Output agreement.** Agreement measures how often decoder-space composition matches latent-space composition. It is computed as the fraction of examples where the two predicted grids match exactly:
$$\mathrm{Agree}_k = \frac{1}{N} \sum_{j=1}^{N} \mathbb{I}\Big[f_\theta\big(f_\theta(x^{(j)}, z_1^{(j)}), z_2^{(j)}\big) = f_\theta\big(x^{(j)}, C_\psi(z_1^{(j)}, z_2^{(j)})\big)\Big].$$
As above, both predictions must match in all 25 cells. Note that this metric does not compare to ground truth; it measures internal consistency between sequential and composed decoding.

**Closure error.** Closure measures how well latent-space composition aligns with decoder-space composition:

$$\text{Closure} = \big\|\text{Prob}\big(f_\theta(f_\theta(x, z_1), z_2)\big) - \text{Prob}\big(f_\theta(x, C_\psi(z_1, z_2))\big)\big\|_2^2.$$

Both sides are converted to per-cell categorical probabilities using a softmax before computing the squared error, matching the implementation.

**Associativity error.**

$$\text{Assoc} = \big\|C_\psi(z_1, C_\psi(z_2, z_3)) - C_\psi(C_\psi(z_1, z_2), z_3)\big\|_2^2.$$

**Identity error.**

$$\text{Id} = \big\|C_\psi(z, z_{\text{id}}) - z\big\|_2^2 + \big\|C_\psi(z_{\text{id}}, z) - z\big\|_2^2.$$

**Latent cosine and MSE.**

$$\text{Cos}(z, \tilde{z}) = \frac{z^\top \tilde{z}}{\|z\| \, \|\tilde{z}\|}, \qquad \text{MSE}(z, \tilde{z}) = \|z - \tilde{z}\|_2^2.$$

## D  COMPOSED LATENTS

For completeness, we clarify how composed latent programs are constructed in our model. Given two single-step transformations represented by pairs $(x, y_1)$ and $(x, y_2)$, the encoder

$$z_i = q_\phi(z \mid x, y_i), \qquad i \in \{1, 2\},$$

infers latent programs corresponding to the individual step-wise transformations.

The composition module then forms a 2-step latent program via

$$z_{\text{comp}} = C_\psi(z_1, z_2),$$

which is intended to represent the transformation obtained by applying the first program followed by the second. To generate a 2-step prediction, the decoder applies this composed program directly to the original input:

$$\hat{y}_{1 \to 2} = f_\theta(x, z_{\text{comp}}).$$

This latent-space composition mirrors the semantic composition of transformations, and the closure and associativity losses enforce consistency between $C_\psi$ and the decoder $f_\theta$. We provide this description here as several reviewers asked for a clearer explanation of how composed latents are constructed in practice.

## E  KL SWEEP ANALYSIS

We conducted a KL sweep over both fixed $\beta$ values and annealed schedules to determine how variational regularization affects the latent program space in **CatLPN**. All runs use 300 training epochs with identical architecture, dataset, and categorical structure. The sweep reveals that *moderate KL, introduced via a short warmup*, provides the best trade-off between compositional performance and latent stability.

**Warmup-20 (main configuration).** The schedule used in the main experiments—a 20-epoch warmup from 0 to $\beta = 0.003$—achieves the strongest overall results:

$$\text{3-step comp. acc.} = 0.8914, \quad \text{3-step output agreement} = 0.8772,$$

with similarly high two-step metrics and a healthy three-step sequential rollout agreement of $0.8274$. This configuration outperforms all other schedules in the sweep by 3–7 points on compositional accuracy and 3–4 points on output agreement, while maintaining strong latent alignment (cosine $\approx 0.962$, MSE $\approx 0.122$).

**Alternative annealing schedules.** A faster warmup (10 epochs to $\beta = 0.003$) and a slower, stronger schedule (30 epochs to $\beta = 0.006$) both perform worse than the 20-epoch schedule: the fast schedule improves over fixed-$\beta$ baselines but lags behind warmup-20 in both 2-step and 3-step metrics, and its three-step sequential agreement is nearly zero; the slow schedule underperforms both warmup-20 and fast annealing across all compositional metrics.

**Fixed KL.** Among fixed KL values, moderate regularization remains preferable to extremes:

- $\beta = 3 \times 10^{-3}$ yields solid two-step and three-step compositional accuracy but does not match either annealed schedule.
- $\beta = 1 \times 10^{-3}$ produces the *best latent alignment* (cosine 0.9696) but noticeably weaker compositional accuracy, showing that high latent similarity does not automatically translate into stable long-horizon behavior.
- $\beta = 0$ achieves reasonable short-horizon metrics but suffers severe drift at longer horizons: three-step compositional accuracy and closure-related metrics collapse.
- $\beta = 10^{-2}$ over-regularizes the latent space and results in the weakest performance across all metrics.

**Takeaway.** KL regularization interacts nonlinearly with categorical structure: too little KL permits latent drift, too much destroys functional capacity, and moderate KL—when introduced via a short warmup—produces the most robust latent geometry for long-horizon composition. Based on these results, we adopt the 20-epoch warmup to $\beta = 0.003$ as the default KL schedule for all main CatLPN experiments.

Table 3: KL sweep results for **CatLPN** (300 epochs). Warmup-20 (20-epoch annealing to $\beta$=0.003) achieves the strongest 2-step and 3-step compositional metrics while maintaining stable three-step rollouts. All runs share identical architecture, dataset, and categorical structure. Metrics: t3_comp = three-step compositional accuracy; t3_oa = three-step output agreement; t2_comp = two-step compositional accuracy; t2_oa = two-step output agreement; t3_oa_three = three-step output agreement based on sequential three-step rollouts.

| Variant | t3_comp | t3_oa | t2_comp | t2_oa | t3_oa_three |
|---|---|---|---|---|---|
| lpn_vae_anneal_w20 | **0.8914** | **0.8772** | **0.8920** | **0.8856** | **0.8274** |
| lpn_vae_anneal_fast | 0.8602 | 0.8444 | 0.8686 | 0.8486 | 0.0026 |
| lpn_vae_anneal_slow | 0.8146 | 0.8090 | 0.8206 | 0.8194 | 0.0006 |
| $\beta$=3 × 10$^{-3}$ | 0.8124 | 0.8094 | 0.8252 | 0.8162 | 0.0000 |
| $\beta$=1 × 10$^{-3}$ | 0.7992 | 0.8212 | 0.8086 | 0.8242 | 0.1398 |
| $\beta$=0 | 0.8138 | 0.8184 | 0.8030 | 0.8160 | 0.0000 |
| $\beta$=10$^{-2}$ | 0.7218 | 0.7402 | 0.7308 | 0.7440 | 0.0000 |

## E.1 Additional Long-Horizon Composition (4-Step)

For completeness, we report 4-step compositional results from a prior 300-epoch full-run configuration (50k/5k split, KL anneal to $0.003$, structure losses enabled). Although our main evaluation focuses on 2-step and 3-step behavior, 4-step composition provides a useful stress test of long-horizon stability.

Despite being trained *only* on single-step examples $(x, y)$, CatLPN maintains strong multi-step performance up to four chained transformations:

$$2\text{-step acc} = 0.8434, \qquad 3\text{-step acc} = 0.8444, \qquad 4\text{-step acc} = 0.8382.$$

Output agreement and closure error also remain stable, indicating that latent-space composition remains aligned with decoder-space composition even at deeper chains.

To better quantify the effect of increased depth, we report performance on the *subsets* that require three- or four-latent compositions within the evaluation set. These subsets measure the model's ability to perform chained composition without relying on intermediate symmetries in the data.

Table 4: Long-horizon evaluation (2–4 steps) for a prior 300-epoch full run with KL anneal to 0.003. Metrics follow the definitions in Appendix C. Performance remains stable at 4 steps, with modest degradation relative to 2-step and 3-step behavior.

| Metric | 2-step | 3-step | 4-step |
|---|---|---|---|
| Compositional Accuracy ↑ | 0.8434 | 0.8444 | 0.8382 |
| Output Agreement ↑ | 0.8342 | 0.8448 | 0.8340 |
| Closure Error ↓ | 0.0012 | 0.0011 | 0.0012 |
| Associativity Error ↓ | 0.0009 | 0.0009 | 0.0009 |
| Identity Error ↓ | 0.0018 | 0.0018 | 0.0018 |
| Latent Cosine ↑ | 0.9599 | 0.9599 | 0.9599 |
| Latent MSE ↓ | 0.1352 | 0.1351 | 0.1351 |

Table 5: Breakdown by depth-specific subsets within the test set. "Three-block" refers to examples requiring three composed transformations; "four-block" refers to examples requiring four composed transformations.

| Subset | Comp. Acc. ↑ | Output Agree. ↑ | Closure Err. ↓ |
|---|---|---|---|
| 3-step (3-block subset) | 0.6280 | 0.5922 | 0.0037 |
| 4-step (3-block subset) | 0.6242 | 0.6036 | 0.0037 |
| 4-step (4-block subset) | 0.5908 | 0.5618 | 0.0041 |

Overall, CatLPN demonstrates stable multi-step behavior beyond the 3-step regime used in the main evaluation. While deeper compositions introduce expected degradation, the latent geometry remains coherent through four composed transformations, supporting the view that categorical constraints promote long-horizon consistency.

## F CATEGORY-THEORETIC FORMALIZATION

We provide a detailed interpretation of our model in categorical terms, using notation consistent with the architectural description.

### F.1 LATENT SPACE AS A CATEGORY

Let $\mathcal{C}$ be a category whose objects correspond to input–output transformation tasks and whose morphisms correspond to latent programs $z \in \mathcal{Z}$. The encoder $q_\phi(z \mid x, y)$ infers such morphisms from transformation pairs $(x, y)$. The composition module $C_\psi$ induces a binary operation

$$z_1 \circ z_2 := C_\psi(z_1, z_2), \qquad z_1, z_2 \in \mathcal{Z}.$$

### F.2 DECODER AS AN APPROXIMATE FUNCTOR

The decoder $f_\theta$ approximately acts as a functor $F : \mathcal{C} \to \mathbf{Set}$, mapping objects to sets of grid states and morphisms to concrete functions $f_\theta(\cdot, z)$ on inputs. Functoriality requires, up to approximation error,

$$f_\theta(x, z_1 \circ z_2) \approx f_\theta(f_\theta(x, z_1), z_2), \qquad f_\theta(x, z_{\mathrm{id}}) \approx x,$$

where $z_{\mathrm{id}} \in \mathcal{Z}$ is the latent identity element.

### F.3 DIFFERENTIABLE CATEGORICAL LAWS

We translate categorical axioms into differentiable losses defined over the latent space $\mathcal{Z}$.

**Associativity.**

$$\mathcal{L}_{\mathrm{assoc}} = \big\| C_\psi(z_1, C_\psi(z_2, z_3)) - C_\psi(C_\psi(z_1, z_2), z_3) \big\|_2^2.$$

**Identity.**

$$\mathcal{L}_{\mathrm{id}} = \big\|C_\psi(z, z_{\mathrm{id}}) - z\big\|_2^2 + \big\|C_\psi(z_{\mathrm{id}}, z) - z\big\|_2^2.$$

**Closure (semantic consistency).**    Categorically, closure requires that composing morphisms in $\mathcal{Z}$ corresponds to composing the induced transformations on data:

$$f_\theta(f_\theta(x, z_1), z_2) \approx f_\theta(x, C_\psi(z_1, z_2)).$$

In practice we enforce this condition through the closure loss $\mathcal{L}_{\mathrm{closure}}$ defined in Appendix A, and we measure how well it holds using the closure error metric described in Appendix C.

### F.4    IMPLICATIONS FOR GENERALIZATION

These losses encourage the latent space to approximate a lawful category: associativity enables systematic multi-step composition, identity stabilizes the latent origin and supports modular reuse, and closure enforces semantic consistency between latent-space composition and decoder-space composition. Empirically, these constraints improve compositional generalization and latent alignment across multi-step tasks.

