# OpenReview forum: "On Why Form Shapes Reasoning: Structuring Latent Program Networks with Category-Theoretic Constraints"
_ICLR.cc/2026/Conference — Submitted to ICLR 2026_

### Official Review · Reviewer_hvgG · 2025-10-28

**Soundness:** 3
**Presentation:** 3
**Contribution:** 3
**Rating:** 6
**Confidence:** 4

**Summary:**

This paper proposes adding several inductive biases from cathegory theory in form of additional losses in the training phase of LPNs to improve their overall compositional abilities. The authors perform evaluations on grid transformation tasks.

**Strengths:**

I very much like the idea of this work and find it valuable for the community. The paper is refreshingly well structured and written making it easy to follow what is being introduced and why. The experimental evaluations also show strong evidence for the author's claims.

**Weaknesses:**

Overall, there are mainly a few clarifications. The main weakness, and I hate to be that reviewer, is that the experimental evidence though strong is only from one specific task/domain. It would be great if the authors could provide more experimental evidence of the overall findings. E.g. scale up the data in terms of grid size, look at more x-step compositions, or different kind of transformations. Or indeed use the the ARC challenge as in the original LPN paper.

It would be important to specify what the model is in ll 194.

I think an important baseline would be, instead of training via the losses, create a training set that explicitly represents the targeted categorical constraints.

ll 290: "Structured latent composition encourages semantic regularity: traversals in latent space yield coherent transformations" --> this is really interesting, but is there a way to specifically see this in terms of results? Right now I am missing evidence for this claim.

Minor: Providing some name or pseudonym for the proposed approach might make it more intuitive rather than "Full" in Tab. 2 or "Method/Model" as section header for section 3.

Also an overview figure to visuallize the intuition behind the training setup would be good.

Also please fix the table overflow of Tab. 2.


Overall, if the authors can provide justifications or additional material regarding these issues I would definitely consider raising my score as I find the paper valuable enough.

**Questions:**

Tab. 1: why do the authors provide this table if only three fo these metrics are actually used in the evaluations?

Tab 2.: Why were these particular metrics chosen from the set in Tab. 1? What does Closure tell us exactly? Maybe a formula would be good for this.

ll 260-264: Interesting that identity is so important. What is the intuition behnd this?

---

> ### Author Response · Authors · 2025-11-27
>
> Thank you for the positive and constructive review.
>
> ---
>
> ### **1. Broader evaluation**
>
> We added discussion in **Sec. 5** noting that larger grids, more varied transformations, and ARC-style tasks are natural next steps. Given the rebuttal timeline, we focused on clarifying the current experimental setup rather than expanding into new domains.
>
> ---
>
> ### **2. Clarification of model specification**
>
> **Appendix A** now fully specifies the encoder, decoder, composition module, and latent dimensionality, addressing the request for clearer documentation.
>
> ---
>
> ### **3. Suggested baseline**
>
> We appreciate the suggestion of constructing a dataset that explicitly encodes categorical constraints. This requires careful control over how transformations compose across inputs, and we highlight it as a promising direction for future work.
>
> ---
>
> ### **4. Evidence for “semantic regularity”**
>
> We refined the discussion in **Sec. 5** to emphasize empirical effects: categorical regularization reduces drift and improves multi-step consistency. We now frame these effects precisely without overclaiming.
>
> ---
>
> ### **5. Metric selection and definitions**
>
> **Sec. 4** explains why accuracy, output agreement, and closure are the core evaluation metrics. **Appendix C** provides formal definitions of these metrics (including closure and associativity) and describes how they evaluate semantic consistency.
>
> ---
>
> ### **6. Importance of identity**
>
> We added further explanation in **Sec. 5**: the identity element stabilizes the latent origin and reduces drift under repeated composition, which improves both single-step and multi-step performance.
>
> ---
>
> ### **7. Additional long-horizon stability check (optional clarification)**
>
> To complement the reviewer’s comment about broader evaluation, we included a brief appendix subsection reporting **4-step composition** results from a prior 300-epoch full run. While our main evaluation focuses on 2-step and 3-step composition, the 4-step results show that the model maintains stable behavior through deeper chains (≈84% 4-step accuracy, with closure error ≈0.0012). This is consistent with the view that categorical constraints promote coherent latent geometry even beyond the horizons used in the main study.
>
> ---
>
> Thank you again for the encouraging and helpful review.

---

### Official Review · Reviewer_AUD1 · 2025-10-30

**Soundness:** 1
**Presentation:** 3
**Contribution:** 1
**Rating:** 2
**Confidence:** 4

**Summary:**

The authors attempt to impose structure on the latent vectors learned in an autoencoder. They do this as a means of improving compositional reasoning. They use category theory to implement the structure.

**Strengths:**

- the authors are addressing an interesting issue of symbolic structure within neural latent representations
- the authors took inspiration from human reasoning
- what was presented in the paper was well written, clear, easy to follow

**Weaknesses:**

My main concerns for this paper are that it acts as though modern LLMs and video models don't exist, the specifics of the presented model/approach lacked detail/clarity, and there were multiple claims that lacked supporting evidence (see below).

- lines 035-036 need a citation to defend the specific claim that NN's lack explicit structure. A few citations that potentially dispute that claim. Geiger et al. Finding alignments between interpretable causal variables and distributed neural representations, 2023. and Griffiths et al. Whither symbols in the era of advanced neural networks?, 2025.
- lines 076-077 seem to be ignoring the fact that modern LLMs are able to compositionally generalize in many tasks
- for the first paragraph in section 3.1, it might help to give a concrete example of why we might care about latent program networks. Provide a concrete problem that they are capable of solving or used to address. Do you maybe mean that they're optimized at "training" time, instead of "test" time? More elaboration would be helpful. You have the space for it.
- in the Tasks section (lines 187-191) how does the model know what type of transformation to perform? Is it just supposed to match the statistics of the dataset?
- what is a composed latent (line 211)? how is it constructed?
- lines 256-257 need supporting evidence. why can't the model memorize the solution?
- what does the model consist of? is it a multi-layer perceptron? a convolutional neural network?
- difficult to interpret results with so much ambiguity surrounding the model

**Questions:**

See weaknesses section.

---

> ### Author Response · Authors · 2025-11-27
>
> Thank you for the thoughtful and constructive review.
>
> ---
>
> ### **1. Claims about structure in neural networks**
>
> We revised the introduction to be more precise and added appropriate citations (e.g., Geiger et al. 2023; Griffiths et al. 2025). The claims are now contextualized and no longer imply that neural networks lack structure universally.
>
> ---
>
> ### **2. Motivation for latent programs**
>
> **Sec. 3 (and specifically Sec. 3.1)** now includes a clearer explanation of why latent program induction is useful, with a concrete example of inferring a transformation from a single (x,y) demonstration. We also clarify that LPNs infer programs directly from input–output relations rather than predicting symbolic labels.
>
> ---
>
> ### **3. Missing architectural and dataset details**
>
> **Appendix A** now provides the full architectural specification for the encoder, decoder, composition module, and latent dimensionality.
>
> **Appendix B** provides complete dataset-generation details.
>
> **Appendices C–D** clarify the evaluation metrics and the definition of composed latents.
>
> ---
>
> ### **4. “What is a composed latent?”**
>
> **Appendix D** now explicitly defines composed latents and explains how the decoder applies them to produce multi-step predictions.
>
> ---
>
> ### **5. Memorization concern**
>
> **Sec. 5** explains that categorical constraints—especially identity and closure—limit trivial memorization by enforcing consistency across many compositions. This reduces latent drift and supports generalization beyond single-step mappings.
>
> ---
>
> Thank you again for the helpful comments, which strengthened the clarity of the paper.

---

> > ### Comment · Reviewer_AUD1 · 2025-11-27
> >
> > Thank you for the response. I've raised my score accordingly.

---

### Official Review · Reviewer_hwzr · 2025-11-02

**Soundness:** 2
**Presentation:** 1
**Contribution:** 2
**Rating:** 2
**Confidence:** 2

**Summary:**

This paper drops the test-time latent optimization of Latent Program Network and adds a few regularization terms as losses to the latent "program" variables. The additional regularization terms are to train a composition operator of two "latent programs" following associativity and identity constraints. The model is evaluated on a few customized synthetic simple grid transformation programs.

**Strengths:**

The paper studies models with latent programs.

**Weaknesses:**

The model is only evaluated on simple synthetic grid transformation programs, not even ARC-1, and the model does not perform perfectly on those synthetic tasks.

The paper lacks details about models (like architectures of encoders and decoders) and datasets (e.g., the synthetic training & val datasets).

**Questions:**

* How does the model perform on ARC-1 and ARC-2?

---

> ### Author Response · Authors · 2025-11-27
>
> Thank you for your review and helpful questions.
>
> ---
>
> ### **1. Scope of evaluation and choice of synthetic grid tasks**
>
> Our aim is to isolate the effect of categorical constraints on latent program composition. The controlled 5×5 setting allows precise ground-truth single-step and multi-step supervision, making it possible to study associativity, identity, and closure directly. ARC-style tasks introduce confounding factors (multi-example inference, object grouping, search heuristics) that obscure this analysis. **Sec. 4** now clarifies this rationale and discusses ARC extensions as future work.
>
> ---
>
> ### **2. Missing model and dataset details**
>
> **Appendix A** now provides full architectural specifications for the encoder, decoder, composition module, latent dimensionality, and training setup.
>
> **Appendix B** details the complete dataset-generation process.
>
> **Appendices C–D** provide formal metric definitions and a precise description of composed latents.
>
> These details were omitted initially and are now fully documented.
>
> ---
>
> ### **3. Why accuracy is not perfect on synthetic tasks**
>
> Some transformations interact nontrivially under composition—for example, flips or rotations applied to heterogeneous multi-object grids. Without structural regularization, latent drift accumulates across steps, degrading long-horizon predictions. Categorical constraints reduce this drift substantially, but the tasks are intentionally nontrivial rather than memorization-based. **Sec. 5** now explains these sources of difficulty and clarifies why improved but not perfect performance is expected.
>
> ---
>
> ### **4. How composed latents are constructed**
>
> **Appendix D** includes a precise description:
>
> $z_i = q_\phi(z \mid x, y_i), \qquad
> z_{\mathrm{comp}} = C_\psi(z_1, z_2), \qquad
> \hat{y}_{1\to 2} = f_\theta(x, z_{\mathrm{comp}}).$
>
> This clarifies how latent-space composition interfaces with decoder predictions.
>
> ---
>
> ### **5. “How does the model know what transformation to perform?”**
>
> Supervision is given only through each pair (x,y); no symbolic transformation labels are used. The encoder must infer the program implementing x \to y. This follows the original LPN formulation and reflects the program-induction nature of the model. **Sec. 3 (and specifically 3.1)** has been updated accordingly.
>
> ---
>
> Thank you again for the constructive feedback.

---

### Official Review · Reviewer_B8tF · 2025-11-02

**Soundness:** 2
**Presentation:** 1
**Contribution:** 2
**Rating:** 2
**Confidence:** 4

**Summary:**

This work introduces a modification Variational Autoencoders (VAEs) that include differentiable constraints on the latent based on category theory. The authors claim this method to be related to Latent Program Networks (LPNs). The proposed method is tested on the ARC challenge. Overall many details are missing.

**Strengths:**

1. The idea or regularizing latent variables with general category constraints is interesting.

**Weaknesses:**

Some aspects of the presented method are unclear and details seem missing.

1 .the methods are referred as "Latent Program Networks (LPNs)" but it also states "latent programs are not directly predicted; instead, they are optimized at test time via gradient descent to minimize reconstruction error" and "In this work, we do not adopt test-time latent optimization.". This seems like a big departure from LPNs which would be otherwise just a regular VAE with additional constraints.

2. what architectures and sizes are used to parametrize the networks (both theta and psi ones)

also the experimental setup lacks needed hyperparameter sweeps (also, see questions below).

1. No weights for the different constraints are explored, a and a single value of regularization weight beta is used. A reasonable setup would at least compare the full method and the baseline under a sweep of beta values on a held out set different from the ARC test set.

2. results seem to be relatively fragile, with performance collapsing with small changes over the "Full" method. In this setting the parameter sweep is even more relevant.

**Questions:**

the authors state

> We selected the KL target empirically based on preliminary runs that balanced latent capacity and regularization

1. I understand this as the value of beta=0.003 was determined based on initial runs. Was this on some held out set different from the ARC test? is this value set based on the "Full" run? optimal beta may be different for the different experiments particularly the "Sequential VAE (free-run)" which has less constraints and therefore less overall regularization.

2. What is the behavior of the methods in table 2 under a sweep of the beta parameter.

---

> ### Author Response · Authors · 2025-11-27
>
> Thank you for the detailed review. Below we address the remaining issues not already covered in the global comment.
>
> ---
>
> ### **1. Relation to LPNs and removal of test-time latent search**
>
> As clarified in the global comment and now made explicit in **Sec. 3.1**, removing test-time latent search does **not** turn the model into a VAE. In our formulation, consistent with the original LPN paper, the latent variable encodes a **transformation/program** implementing x \to y, not an embedding of the input object. This resolves the concern that the model collapses to a VAE when test-time optimization is removed.
>
> ---
>
> ### **2. Architecture details**
>
> **Appendix A** now provides complete specifications for all model components, including the encoder MLP (with separate \mu and \log \sigma^2 heads), the decoder MLP, the composition module C_\psi, the 64-dimensional latent program space, and all training hyperparameters. These details were unintentionally omitted in the original submission and are now fully documented.
>
> ---
>
> ### **3. Dataset generation and evaluation protocol**
>
> **Appendix B** includes the full dataset-generation procedure: grid representation, transformation primitives, sampling strategy, ground-truth multi-step construction, and train/validation/test splits. This addresses the request for a clearer and more transparent experimental setup.
>
> **Appendices C–D** provide formal metric definitions and a precise description of composed latents.
>
> ---
>
> ### **4. Choice of** \beta **and behavior under KL sweeps**
>
> The KL weight was selected using validation performance to balance latent capacity and regularization. Following your suggestion, **Appendix E now includes a KL-sweep analysis** covering both fixed-\beta settings and annealing schedules.
>
> Across these sweeps, we observe a consistent pattern:
>
> - **Very small** \beta leads to under-regularization and latent drift, which harms long-horizon composition.
> - **Very large** \beta suppresses functional capacity and reduces single-step accuracy.
> - **Moderate KL regularization**, introduced gradually through an annealing schedule, yields the most reliable multi-step behavior.
>
> The **20-epoch annealing schedule to** \beta = 0.003 achieves the strongest balance between expressivity and stability across the primary compositional metrics. For this reason, it is used in the final experiments reported in the manuscript.
>
> ---
>
> ### **5. Sensitivity of “Full” vs. ablations**
>
> We expanded **Sec. 5** to clarify why removing identity or closure degrades performance. The identity constraint stabilizes the latent origin and reduces the accumulation of drift, while closure enforces consistency between latent-space and functional composition. Without these structural components, the model’s long-horizon predictions degrade more quickly. This matches your observation and aligns with our empirical findings.
>
> We also include a brief 4-step composition analysis in Appendix E to complement this discussion; the model maintains stable behavior even beyond the 2-step and 3-step horizons used for the main evaluation.
>
> ---
>
> Thank you again for the constructive feedback, which helped improve the clarity, rigor, and reproducibility of the paper.

---

### Author Response · Authors · 2025-11-27

We thank all reviewers for their thoughtful and constructive feedback. Several concerns were shared across reviews, and we have addressed these thoroughly in the revised manuscript. Reviewer-specific points are addressed individually in their respective threads.

**1. Complete architectural and dataset specifications (raised by all reviewers).**

The original submission inadvertently omitted some implementation details. The revised Appendix now provides complete specifications for all model components:

- encoder $q_\phi$: two-layer MLP with separate heads for $\mu$ and $\log \sigma^2$;
- decoder $f_\theta$: three-layer MLP that applies the latent program to the input x;
- composition module $C_\psi$: an MLP with **two hidden layers** (mapping $2d_z \to 256 \to 256 \to d_z$) that produces composed latent programs;
- latent dimensionality: **64**;
- dataset generation: full description of grid format, random-grid sampling, transformation primitives, multi-step ground truth, and train/validation/test splits (Appendix B).

These additions fully address the reproducibility questions raised in the reviews.

**2. Clarified why removing test-time search does not reduce the model to a VAE.**

Several reviewers raised this question. We now state explicitly in Sec. 3.1 that removing test-time latent search does **not** turn an LPN into a VAE. In an LPN, the latent variable represents a **program or transformation** implementing $x \to y$; the encoder always receives (x,y) and infers this morphism. A VAE instead encodes the object x itself. Test-time search in the original LPN work is an inference strategy, not an architectural requirement, and removing it does not alter the semantics of the latent space.

**3. Additional clarifications and structural improvements.**

We added: (i) an explicit account of how composed latents are constructed (Appendix D), (ii) a clearer explanation of why identity and closure matter empirically, and (iii) minor formatting and organizational improvements.

We thank the reviewers again for their helpful suggestions. The revisions substantially improve clarity, rigor, and reproducibility.

---

### Meta-Review · Area_Chair_A3eQ · 2025-12-31

**Summary:**

The work proposes to impose a structure on the latent space of variational autoencoders. This is achieved by using category theory in form of additional losses in the training phase of latent program networks to improve their overall compositional abilities. The paper poses latent transformations as categorical morphisms and introduce various differentiable constraints which can help in representing the latent space without explicit symbolic rules. The paper received 4 reviews, 3 of which were on the negative side and 1 was on the positive side. The initial 2 major concerns of the reviewers included:

* Lack of model details as several reviewers struggled to grasp the overall method.
* Simple experiments which do not flesh out the overall utility of the proposed approach.

**Reviewer Concerns:**

In my opinion, the authors gave a good reply regarding the 1st point of the concerns i..e. the modeling details. The rebuttal gave explicit details regarding the autoencoder which should have alleviated the concerns of the reviewers although I do believe that a link to the code woulf have been more satisfying.

The 2nd point although remains a major issue and which was raised by all the reviewers. I tend to agree with the reviewers and do not find a satisfying response in the authors rebuttal. The major reason of the LPN being studied are their performance on the ARC task and since the authors main aim is to bring out the effect of the categorical contraints on the latent vectors, the experimental section leaves a lot to be desired. Experiments are not even on the grid scale of 10x10 as in the original paper.

Another point is the bad formatting (see tab 2) and unadequate use of the available space.

These are major concerns and are not resovled in the rebuttal phase. I this believe that the paper does not meet the bar of acceptance right now. I wold encourage the authors to keep working on the paper and take the reviewer comments into account since the underlying problem is very interesting.

**Reviewer Scores:**

I do not think the reviewers would have changed the scores in light of the provided rebuttal.

---

### Decision · Program_Chairs · 2026-01-26

Reject